# β-Catenin regulates endocardial cushion growth by suppressing p21

Huahua Liu[1],*, Pengfei Lu[2],*, Shan He[3], Yuru Luo[3], Yuan Fang[3], Sonia Benkaci[2], Bingruo Wu[2], Yidong Wang[4] , Bin Zhou[5]

Endocardial cushion formation is essential for heart valve development and heart chamber separation. Abnormal endocardial cushion formation often causes congenital heart defects. β-Catenin is known to be essential for endocardial cushion formation; however, the underlying cellular and molecular mechanisms remain incompletely understood. Here, we show that endothelial-specific deletion of β-catenin in mice resulted in formation of hypoplastic endocardial cushions due to reduced cell proliferation and impaired cell migration. By using a β-catenin[DM] allele in which the transcriptional function of β-catenin is selectively disrupted, we further reveal that β-catenin regulated cell proliferation and migration through its transcriptional and non-transcriptional function, respectively. At the molecular level, loss of β-catenin resulted in increased expression of cell cycle inhibitor p21 in cushion endocardial and mesenchymal cells in vivo. In vitro rescue experiments with HUVECs and pig aortic valve interstitial cells confirmed that β-catenin promoted cell proliferation by suppressing p21. In addition, one savvy negative observation is that β-catenin was dispensable for endocardial-to-mesenchymal fate change. Taken together, our findings demonstrate that β-catenin is essential for cell proliferation and migration but dispensable for endocardial cells to gain mesenchymal fate during endocardial cushion formation. Mechanistically, β-catenin promotes cell proliferation by suppressing p21. These findings inform the potential role of β-catenin in the etiology of congenital heart defects.

## Introduction

Endocardial cushions give rise to heart valves and the septa that separate the cardiac chambers (Lin et al, 2012) and maintain the unidirectional blood flow in the heart, which are essential for the normal heart function (Wu et al, 2017; O'Donnell & Yutzey, 2020). In mice, the endocardial cushions begin to form within atrioventricular canal (AVC) and outflow tract (OFT) through endocardial to mesenchymal transformation (EndoMT) between E9.5 and E10.5 (Hinton & Yutzey, 2011). These cushions continue to grow through cell proliferation between E10.5 and E12.5. After E12.5, the endocardial cushions undergo a complex remodeling process that involves cell proliferation and apoptosis and extracellular matrix organization and generate the thin heart valve leaflets and the membranous septum structures at birth (Combs & Yutzey, 2009; Tao et al, 2012). Endocardial cushion formation is precisely controlled by signals from endocardium and myocardium (Armstrong & Bischoff, 2004; Combs & Yutzey, 2009). Dysregulation of these signals can cause endocardial cushion malformations, leading to congenital heart defects (Lincoln & Yutzey, 2011; Dutta et al, 2021). Therefore, a deeper understanding of the cellular and molecular mechanisms controlling endocardial cushion formation is critically needed for identifying the molecular etiology of congenital heart defects and for better management of these diseases.

β-Catenin is encoded by Ctnnb1 gene and conducts two distinct functions at cell membrane and nucleus, respectively (Gessert & Kuhl, 2010). It interacts with VE-cadherin at cell membrane and maintains cell–cell adhesion. On the other hand, β-catenin mediates the canonical WNT signaling by forming a transcriptional complex with TCF/LEF in cell nucleus (MacDonald et al, 2009). In mouse, canonical WNT signaling plays essential roles in heart valve development including endocardial cushion formation, and its overactivation is associated with calcific aortic valve disease (Alfieri et al, 2010; Askevold et al, 2012; Gu et al, 2014; Albanese et al, 2017). Increased WNT/β-catenin signaling causes developmental heart valve malformation and leads to adult heart valve disease (Xu & Gotlieb, 2013; Thalji et al, 2015; Hulin et al, 2017). WNT/β-catenin signaling promotes mesenchymal cell proliferation essential for endocardial cushion growth through both cell autonomous and non-autonomous mechanisms (Cai et al, 2013; Wang et al, 2018). A KLF2–WNT9b paracrine signaling axis senses the flow and directs

[1]Department of Cardiology, First Affiliated Hospital, Xi'an Jiaotong University, Xi'an, China   [2]Department of Genetics, Albert Einstein College of Medicine, Bronx, NY, USA   [3]The Institute of Cardiovascular Sciences, School of Basic Medical Sciences, Xi'an Jiaotong University, Xi'an, China   [4]The Institute of Cardiovascular Sciences, School of Basic Medical Sciences; Department of Cardiology, First Affiliated Hospital; Key Laboratory of Environment and Genes Related to Diseases of Ministry of Education, Xi'an Jiaotong University, Xi'an, China   [5]Departments of Genetics, Pediatrics (Pediatric Genetic Medicine), and Medicine (Cardiology), The Wilf Family Cardiovascular Research Institute, The Einstein Institute for Aging Research, Albert Einstein College of Medicine, Bronx, NY, USA

Correspondence: yidwang119@xjtu.edu.cn; bin.zhou@einsteinmed.edu
*Huahua Liu and Pengfei Lu contributed equally to this work

the proper cushion remodeling into mature heart valve leaflets (Goddard et al, 2017). In zebrafish, activation and inhibition of the canonical WNT signaling promotes and represses endocardial cushion formation, respectively (Hurlstone et al, 2003). Similarly, in avian, overexpression of WNT9a leads to excessive canonical WNT signaling and increased endocardial cushion size (Person et al, 2005). In mice, a NOTCH–WNT–BMP signal axis between endocardium and myocardium has been reported to control EndoMT and endocardial cushion formation during early heart valve development (Wang et al, 2013). Early studies by Liebner et al reported that targeted deletion of β-catenin in the endothelium using $Tie2^{Cre}$ driver disrupts EndoMT and results in the formation of hypocellular endocardial cushions (Liebner et al, 2004), whereas the molecular and cellular mechanisms remain to be determined. In contrast, recent studies by Bosada et al showed that inhibition of canonical WNT signaling by overexpressing DKK1 in the endocardium had no effect on EndoMT and AVC cushion formation (Bosada et al, 2016). These two studies suggest that β-catenin in the endocardium regulates the EndoMT independent of its canonical WNT signaling function. This idea is supported by the fact that canonical WNT activities are high in the myocardium and absent in the endocardium at E9.5 when EndoMT begins (Wang et al, 2018). Therefore, further studies are needed to determine the cellular and molecular mechanisms by which β-catenin in the endocardium regulates EndoMT and subsequent endocardial cushion growth.

This study addresses this knowledge gap by complete abolishment of β-catenin functions or selective disruption of the transcriptional function of β-catenin in endocardium. Our results show that the transcriptional and non-transcriptional function of β-catenin regulates cell proliferation and migration, respectively, during endocardial cushion formation. Loss of β-catenin results in defective filopodia formation and cell migration. The mechanism behind the cell proliferation defect involves up-regulation of p21. Together, these findings provide new insights into the cellular and molecular mechanisms through which β-catenin controls endocardial cushion formation.

# Results

## β-Catenin promotes cell proliferation and migration during endocardial cushion formation

We crossed floxed β-catenin mice with $Tie2^{Cre}$ mice to generate the endothelial-specific β-catenin knockout mice ($β$-catenin$^{f/f}$:Tie2$^{Cre}$, referred as $β$-cat$^{eKO}$ hereafter). Immunostaining confirmed that β-catenin was expressed in both endocardium and myocardium of E9.5 control ($β$-catenin$^{f/f}$ or $β$-catenin$^{f/+}$) embryos, but its expression in endocardium was selectively disrupted in their littermate $β$-cat$^{eKO}$ embryos (Fig 1A). $β$-cat$^{eKO}$ embryos were underdeveloped compared with their littermate controls at E10.5, although they were grossly normal and indistinguishable from controls at E9.5 (Fig S1). HE staining showed that AVC cushions of E9.5 control embryos were being populated by mesenchymal cells, indicating proper formation of endocardial cushions (Fig 1B). In contrast, AVC cushions of $β$-cat$^{eKO}$ embryos were hypocellular with fewer mesenchymal cells

(Fig 1B and C). $Tie2^{Cre}$ also deletes genes at the OFT region where EndoMT starts around E9.5, slightly later than the beginning time of EndoMT in the AVC region (Kisanuki et al, 2001). Indeed, we confirmed that there were very few, if any, mesenchymal cells within OFT cushions of either control or $β$-cat$^{eKO}$ embryos at E9.5 (Fig S2). EdU labeling showed significantly reduced cell proliferation in endocardium, mesenchyme, and myocardium of $β$-cat$^{eKO}$ embryos compared with controls (Fig 1D and E). By comparing the histology of AVC cushions between control and $β$-cat$^{eKO}$ embryos, we found that the mesenchymal cells in cushions of control embryos migrated far away from the endocardium and invaded into matrix-rich cushions (Fig 1B). In contrast, mesenchymal cells in the cushion of $β$-cat$^{eKO}$ embryos failed to migrate away from the endocardium; instead, they were clustered underneath the endocardium (Fig 1B). Quantification of the cell distribution showed that 40% of mesenchymal cells in control embryos were located within the sub-endocardium region, although this number was greatly increased to 80% in $β$-cat$^{eKO}$ embryos (Fig 1F), indicating a migratory defect.

We then characterized this migratory defect by performing an EndoMT assay on collagen gel using AVC tissue explants isolated from hearts of E9.5 embryos. Consistent with in vivo results, EndoMT assay showed that fewer cells migrated out from the explants of $β$-cat$^{eKO}$ embryos when compared with that of control explants (Fig 2A and B). Live-cell tracking showed that mesenchymal cells of control explants migrated far away from the explants whereas those cells of $β$-cat$^{eKO}$ explants stayed around the explants (Fig 2C). In addition, mesenchymal cells of $β$-cat$^{eKO}$ explants exhibited significantly reduced migration velocity and mean square displacement (Fig 2D and E), further suggesting a migratory defect. In contrast, the migration directionality of mesenchymal cells was comparable between control and $β$-cat$^{eKO}$ explants (Fig 2F). These findings support that β-catenin is required for cell proliferation and migration during endocardial cushion formation.

## β-Catenin is required for filopodia formation

β-Catenin forms a complex with adenomatous polyposis coli at membrane protrusions and modulate mammary tumor cell migration and mesenchymal phenotype (Odenwald et al, 2013). N-terminally phosphorylated β-catenin (phospho-β-catenin) was shown to accumulate at the leading edge of migrating cells (Faux et al, 2010). Consistently, we found that phospho-β-catenin (Thr41/Ser45) was highly expressed in the protrusions (filopodia/lamellipodia) of transforming endocardial cells and transformed mesenchymal cells in control embryos, although such expression was dramatically disrupted in $β$-cat$^{eKO}$ embryos (Fig 3A and B). The expression pattern of β-catenin suggests that it may be involved in regulation of filopodia formation. To test this hypothesis, we performed whole-mount staining of isolectin B4 (IB4) and αSMA. The results showed that endocardial and mesenchymal cells at the AVC cushions of E9.5 control embryos exhibited prominent protrusions toward the myocardium (Fig 3C and Video 1). In contrast, very few cells in AVC of $β$-cat$^{eKO}$ embryos presented such protrusions (Fig 3C and Video 2). Consistently, staining of F-actin with phalloidin showed that endocardial and mesenchymal cells in AVC of E9.5 control embryos formed microspikes (filopodia), whereas very few cells in AVC of $β$-cat$^{eKO}$

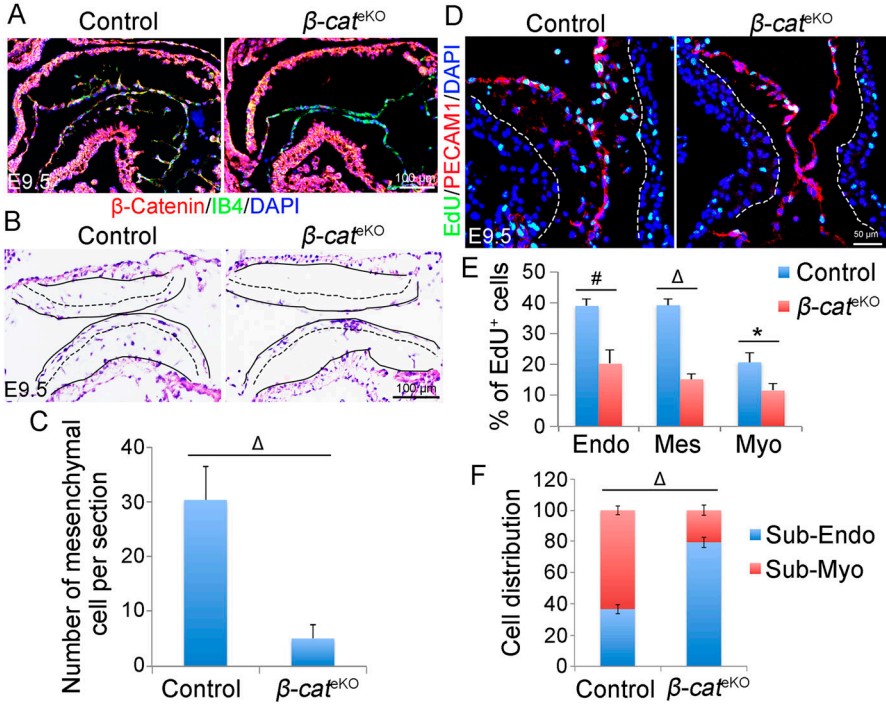

**Figure 1. Deletion of β-catenin in endocardium disrupts endocardial cushion formation.**
**(A)** Immunostaining on heart sections from E9.5 control (*β-cat^{f/f}* or *β-cat^{f/+}*) and *β-cat^{eKO}* (*β-cat^{f/f}:Tie2^{Cre}*) embryos with *β*-catenin antibody (red) and isolectin B4 (IB4) was co-stained to mark the endocardium (green). **(B)** H&E staining of E9.5 embryo sections. **(C)** Quantification of mesenchymal cell number within atrioventricular cushions. n = 5/group. **(D)** EdU labels proliferating cells and immunostaining of PECAM1 marks the endocardium. **(E)** Quantification of proliferating endocardial (Endo), mesenchymal (Mes), and myocardial (Myo) cells. n = 3/group. **(B, F)** Quantitative analysis of cell distribution showed in (B) shows the percentage of mesenchymal cells near (sub-Endo) or far away from (sub-Myo) the endocardium. n = 4/group. # < 0.01; Δ < 0.001 by unpaired *t* test.

embryos had this feature (Fig 3D). Together, these findings indicate that *β*-catenin is essential for filopodia formation during endocardial cushion formation.

### β-Catenin regulates cell proliferation transcriptionally and migration non-transcriptionally

*β*-Catenin is a dual functional protein–mediating cell–cell adhesion at cell membrane and transcriptional regulation in the nucleus (Valenta et al, 2011). We used a *β-catenin^{DM}* allele to disrupt selectively the transcriptional function of *β*-catenin (Valenta et al, 2011) and generated a compound *β*-catenin mutant mice (*β-cat-enin^{f/DM}:Tie2^{Cre}*, referred as *β-cat^{eKO/DM}* hereafter) in which endocardial cells and their mesenchymal progenies completely lost the transcriptional function of *β*-catenin, whereas maintained one copy of *β*-catenin with the cell adhesive function. We then analyzed the AVC cushion development in control, *β-cat^{eKO/eKO}* (disruption of both functions of *β*-catenin in endocardial cells and their progenies) and *β-cat^{eKO/DM}* embryos. HE staining showed that both *β-cat^{eKO/eKO}* and *β-cat^{eKO/DM}* embryos had hypocellular AVC cushions (Fig 4A and B) and decreased cell proliferation (Fig 4D and E), indicating that transcriptional function of *β*-catenin regulates cell proliferation. In contrast, many mesenchymal cells in AVC cushions of control and *β-cat^{eKO/DM}* embryos migrated far away from the endocardium, whereas those cells of *β-cat^{eKO/eKO}* embryos failed to do so (Fig 4A and C). Consistently, whole-mount staining of IB4 showed that mesenchymal cells in control and *β-cat^{eKO/DM}* embryos had prominent filopodia, whereas few, if any, cells in *β-cat^{eKO/eKO}* embryos acquired this structure (Fig 4F and Video 3, Video 4, and Video 5), confirming that the non-transcriptional function of *β*-catenin regulates filopodia formation and cell

migration. These findings support that transcriptional and non-transcriptional function of *β*-catenin promotes cell proliferation and migration, respectively, during endocardial cushion formation.

### β-Catenin supports cell proliferation by suppressing p21

To determine the molecular mediators through which *β*-catenin regulates cell proliferation and migration, we examined the expression of candidate genes that are known to be involved in early endocardial cushion formation. The AVC tissues were microdissected from E9.5 control, *β-cat^{eKO/eKO}*, and *β-cat^{eKO/DM}* embryos and used for gene expression analysis by quantitative PCR (qRT-PCR). The qRT-PCR results showed that the expression of *Notch1*, *Hey1*, *Hey2*, *Bmp2*, *Tgfbr2*, *Bmpr1a*, and *Bmpr2*, which are involved in the NOTCH and BMP signals essential for EndoMT (Garside et al, 2013; Wang et al, 2021), was not affected in either *β-cat^{eKO/eKO}* or *β-cat^{eKO/DM}* embryos (Fig S3). We confirmed by immunostaining that active NOTCH1 (N1ICD) and BMP downstream effector pSMAD1/5/9 were not affected in both mutant embryos (Fig S4). The expression of *Nrg1*, which regulates EndoMT during endocardial cushion formation through the NRG1/ErBB pathway (Armstrong & Bischoff, 2004), was not affected in *β-cat^{eKO/eKO}* embryos but significantly upregulated in *β-cat^{eKO/DM}* embryos (Fig S3). The expression of *Erbb3* was significantly reduced in both mutant embryos, whereas the expression of *Erbb2* and *Erbb4* was not changed in either group (Fig S3). In addition, the ID genes (*Id1*, *Id2*, and *Id3*), known for their functions in EndoMT (Fraidenraich et al, 2004; Kowanetz et al, 2004), were not affected in either *β-cat^{eKO/eKO}* or *β-cat^{eKO/DM}* embryos (Fig S3). As matrix protein dynamics are essential for endocardial cushion formation (Hinton et al, 2006), we also examined the expression of matrix genes (*Has2*, *Acan*, and *Vcan*) and found their

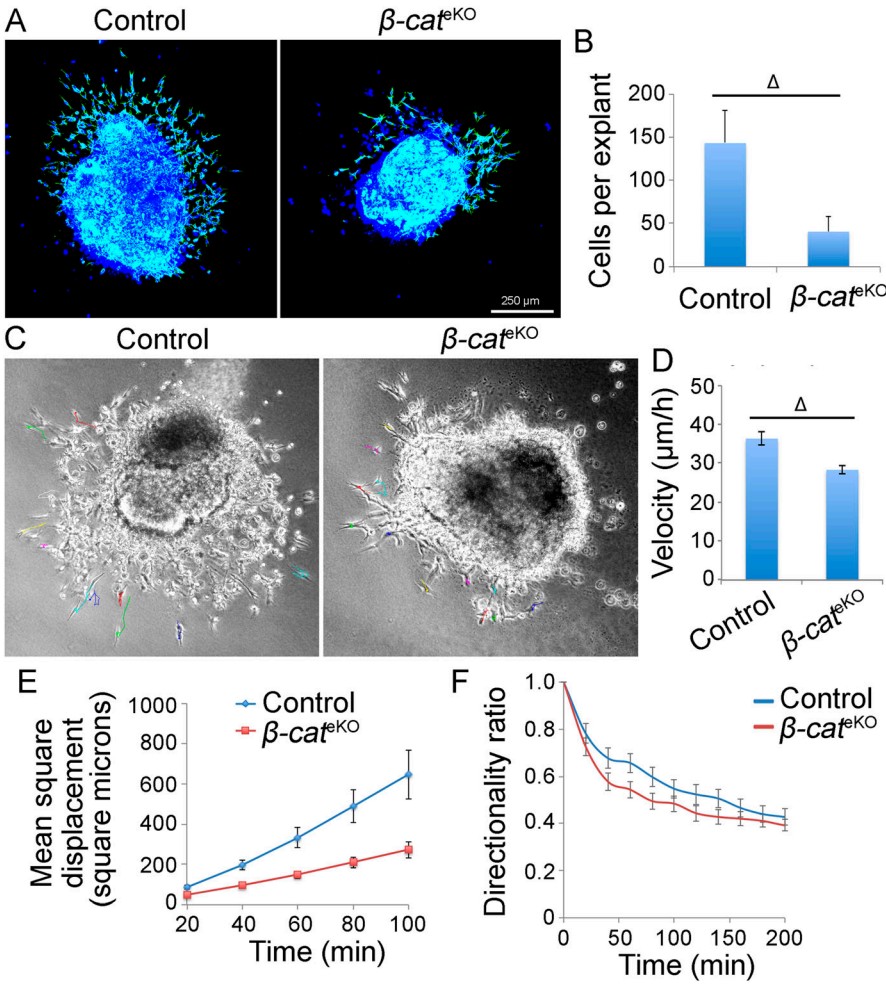

**Figure 2. β-catenin loss impedes mesenchymal cell migration.**
**(A)** AVC tissues from E9.5 embryos were cultured in collagen gel for 48 h. The explants were stained with phalloidin for F-actin. **(B)** Bar graph shows the number of cells that migrated into the gel (B). **(C, D, E, F)** and cell tracking analysis of time-lapse images from the AVC explants shows velocity (D), mean square displacement (E), and directionality (F) of the migratory cells using DiPer. Unpaired *t* test was used for the statistical calculation. Δ < 0.001.

expression was not altered in either mutant (Fig S3). In contrast, the qRT-PCR results showed that the expression of *Snail*, *Slug*, *Twist1*, and *Msx1*, which critically control EndoMT during endocardial cushion formation (Wirrig & Yutzey, 2011), was markedly reduced in both mutant embryos, whereas the expression of *Msx2* was not changed in either mutant (Fig S3).

Consistent with the phenotype of reduced proliferation, cell cycle inhibitor p21 was dramatically increased in both β-cat[eKO/eKO] and β-cat[eKO/DM] embryos (Fig 5A). RNAscope and immunostaining confirmed that the number of p21-expressing endocardial or mesenchymal cells in the AVC region was dramatically increased in both mutant embryos when compared with that in control embryos (Fig 5B–D). In line with the in vivo findings, β-catenin inhibition by XAV939 significantly up-regulated the expression of p21 in HUVECs and pig aortic valve interstitial cells (PAVIC) (Fig 5E and F). Together, these results indicate that β-catenin promote cell proliferation likely by suppressing p21 expression.

We then performed rescue experiments to determine whether the elevated p21 expression contributed to the cell proliferation defect caused by β-catenin deficiency. β-Catenin knockdown dramatically up-regulated p21 expression and inhibited cell proliferation in HUVEC, and p21 knockdown rescued the cell proliferation defect

(Fig 6A and C). Similarly, β-catenin inhibition with XAV939 markedly repressed the proliferation of PAVIC, which was partially restored by p21 inhibition with UC2288 (Fig 6B and D). Taken together, these results demonstrate that p21 suppression by β-catenin is essential for cell proliferation and proper endocardial cushion formation.

## β-Catenin is dispensable for endocardial-to-mesenchymal fate change

Endocardial cushions formation begins with an EndoMT process in which some cushion endocardial cells gradually loose the endothelial marker VE-cadherin and gain the mesenchymal markers (αSMA, VIMENTIN, PDGFRβ) (Armstrong & Bischoff, 2004). Down-regulation of VE-cadherin is prerequisite for transformed endocardial cells to delaminate from the endocardial sheet and migrate into the matrix-rich cushions (Timmerman et al, 2004). Immunostaining showed that VE-cadherin protein in AVC endocardium was significantly increased in β-cat[eKO/eKO] embryos but not in β-cat[eKO/DM] embryos when compared with that in controls, although it in ventricular endocardium was comparable among three groups (Fig S5A). This result indicates that β-catenin negatively regulates the protein level of VE-cadherin via its non-transcriptional function. In support of

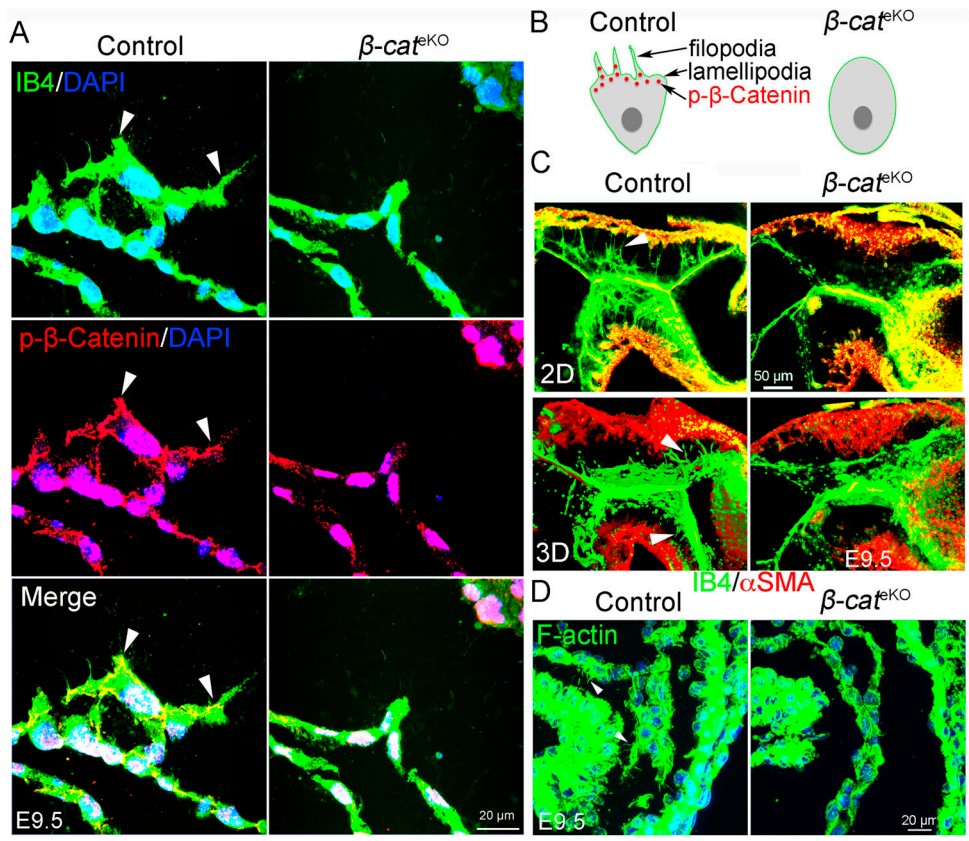

**Figure 3. β-catenin loss disrupts filopodia formation.**
**(A)** Immunostaining of phospho-β-catenin (red) and IB4 on tissue sections of E9.5 embryos. IB4 staining (green) shows the transforming endocardial cells within the AVC region of control embryos having well-formed membrane protrusions with enriched phospho-β-catenin (arrowhead), whereas the β-cat^eKO embryos lack these features. **(B)** A carton depicts the expression pattern of phospho-β-catenin. **(C)** Representative images from whole mount staining of E9.5 embryos with αSMA (red) and IB4 (green) antibodies show migrating mesenchymal cells within AVC region of control embryos possessing prominent filopodia (arrowhead), whereas these structures are absent in the β-cat^eKO embryos. **(D)** Representative images from phalloidin staining for F-actin show migrating mesenchymal cells in control embryos having distinct microspikes (arrowhead), whereas these structures are not present in the β-cat^eKO embryos.

this notion, RNAscope and qRT-PCR results showed that the mRNA level of *VE-cadherin* in cushion endocardial cells was comparable among control, β-cat^eKO/eKO, and β-cat^eKO/DM embryos (Fig S5B and C). On the other hand, immunostaining showed that the cushion endocardial cells in control, β-cat^eKO/eKO, and β-cat^eKO/DM embryos expressed a comparable level of mesenchymal markers including αSMA, VIMENTIN, and PDGFRβ (Fig 7A). SNAIL, SLUG, and SOX9 are key transcriptional factors inducing endocardial cells to gain mesenchymal fate. The expression of these markers in cushion endocardium was not affected by β-catenin loss (Fig 7B). Together, these results indicate that β-catenin is dispensable for endocardial cells to gain mesenchymal fate.

## Discussion

Although β-catenin is known to be essential for endocardial cushion formation during heart valve development, the underlying cellular and molecular mechanisms remain elusive. In the present study, we addressed this question by using mouse models and cell cultures. Our findings demonstrate that β-catenin is essential for cell proliferation and migration during endocardial cushion formation. Moreover, we show that β-catenin regulates cell proliferation and migration via its transcriptional and non-transcriptional function, respectively. Mechanistically, β-catenin promotes cell proliferation by suppressing *p21* expression and supports cell migration via mediating filopodia formation (Fig 8A and B).

Heart valve development begins with the endocardial cushion formation through a process involving cell differentiation (EndoMT), proliferation, and migration. WNT/β-catenin signaling is essential for endocardial cushion formation in zebrafish and mice. Consistent with previous reports, we found that endothelial-specific deletion of β-catenin in mice resulted in hypoplastic endocardial cushions. In zebrafish, truncated adenomatous polyposis coli–mediated activation of WNT/β-catenin signaling led to excessive endocardial cushions associated with increased cell proliferation (Hurlstone et al, 2003). In addition, overactivation of WNT/β-catenin induced ectopic expression of *notch1b* whose expression was normally specific in the valve endocardial cells in zebrafish (Hurlstone et al, 2003). These findings suggest that WNT/β-catenin is essential for endocardial cell fate determination and cell proliferation during endocardial cushion formation in zebrafish. In mice, Liebner et al showed that endocardial cells with deletion of β-catenin failed to transdifferentiate into mesenchymal cells in an ex vivo AV explant assay, indicating β-catenin is required for the endocardial-to-mesenchymal fate change (Liebner et al, 2004). In this study, cell proliferation was not investigated in vivo, whereas in vitro experiments revealed that β-catenin loss had no effect on cell proliferation. In the present study, we further investigated cellular and molecular mechanisms through which β-catenin regulates endocardial cushion formation. In contrast, we showed by immunostaining that the cushion endocardial cells in control and β-catenin knockout embryos expressed similar levels of mesenchymal markers (αSMA, VIMENTIN, PDGFRβ) and transcriptional factors essential

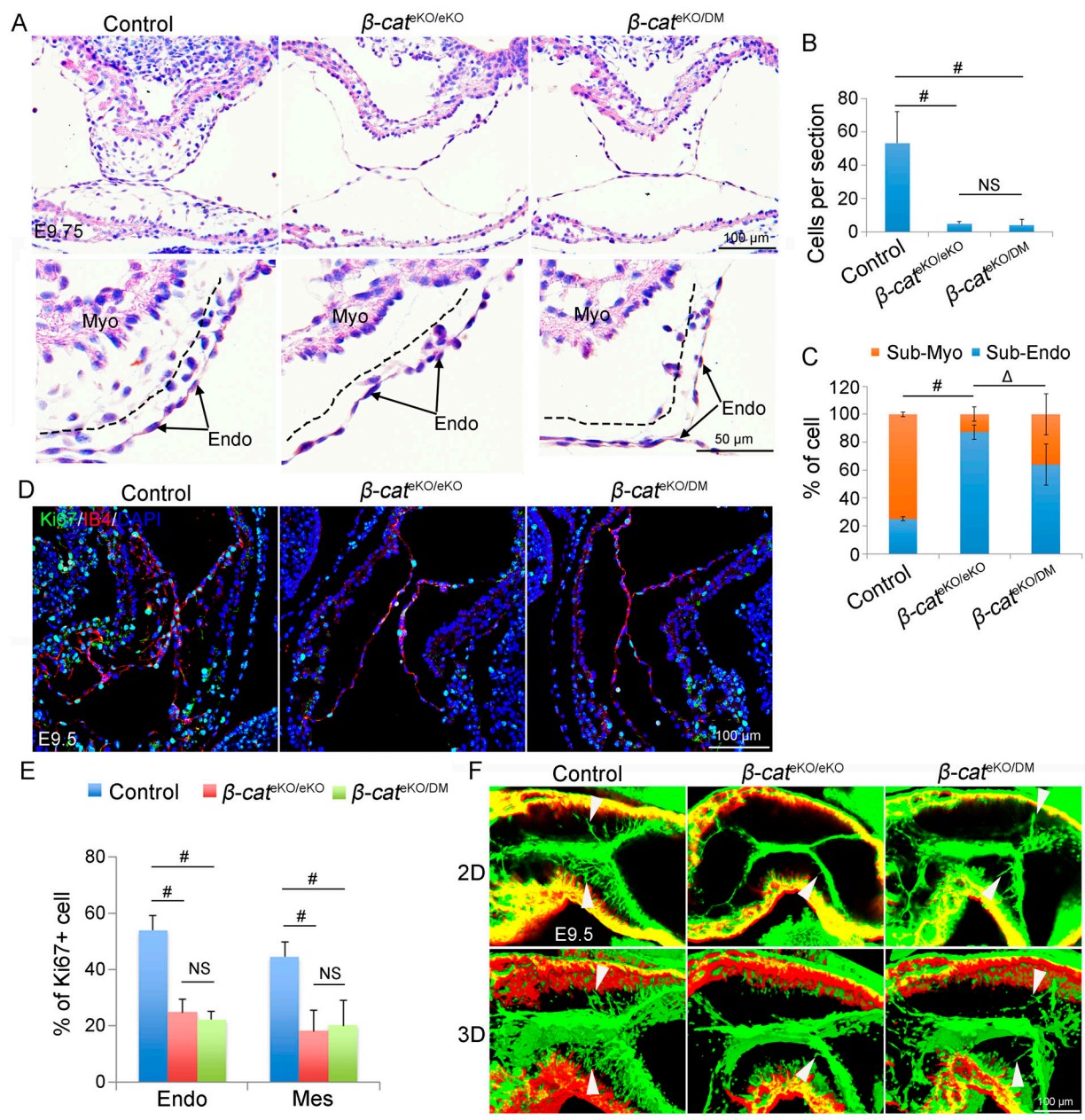

**Figure 4. β-catenin regulates cell proliferation and migration through its transcriptional and non-transcriptional function, respectively.**
**(A)** H&E staining shows AVC morphology of E9.5 control (β-cat^f/f or β-cat^f/+), β-cat^eKO/eKO (β-cat^f/f:Tie2^Cre), and β-cat^eKO/DM (β-cat^f/DM:Tie2^Cre) embryos. β-catenin^DM allele was used to separate the transcriptional and non-transcriptional functions of β-catenin. **(B)** Quantification of mesenchymal cell number within atrioventricular cushions. n = 4/group. **(A, C)** Quantitative analysis of cell distribution showed in (A) shows the percentage of mesenchymal cells near (sub-Endo) or away from (sub-Myo) the endocardium. n = 4/group. **(D)** Ki67 antibody staining labels proliferating cells (green), and IB4 immunostaining marks the endocardium (red). **(E)** Quantification of proliferating rate of endocardial (Endo) and mesenchymal (Mes) cell. n = 5/group. One-way ANOVA was used for the statistical calculation. # < 0.01; Δ < 0.001. **(F)** Representative images of whole-mount staining of E9.5 embryos with αSMA and IB4 antibodies show filopodia in the migrating mesenchymal cells within AVC region of the control and β-cat^eKO/DM embryos (arrowhead), whereas they are not present in β-cat^eKO/eKO embryos.

for EndoMT, suggesting that β-catenin is dispensable for endocardial cells to gain mesenchymal fate. The inconsistent role of β-catenin in cell fate determination revealed by different studies may be due to different markers and methods been used. In agreement with the findings in zebrafish, we showed by EdU labeling that β-catenin is essential for cushion endocardial and mesenchymal cell

proliferation during early endocardial cushion formation. Consistently, we have reported that β-catenin promotes post-EMT cushion growth via both cell autonomous and non-autonomous mechanisms (Wang et al, 2018). In addition, by carefully analyzing the cushion morphology and tracing live cells in AVC explants, we identified a previously undefined function for β-catenin in filopodia formation

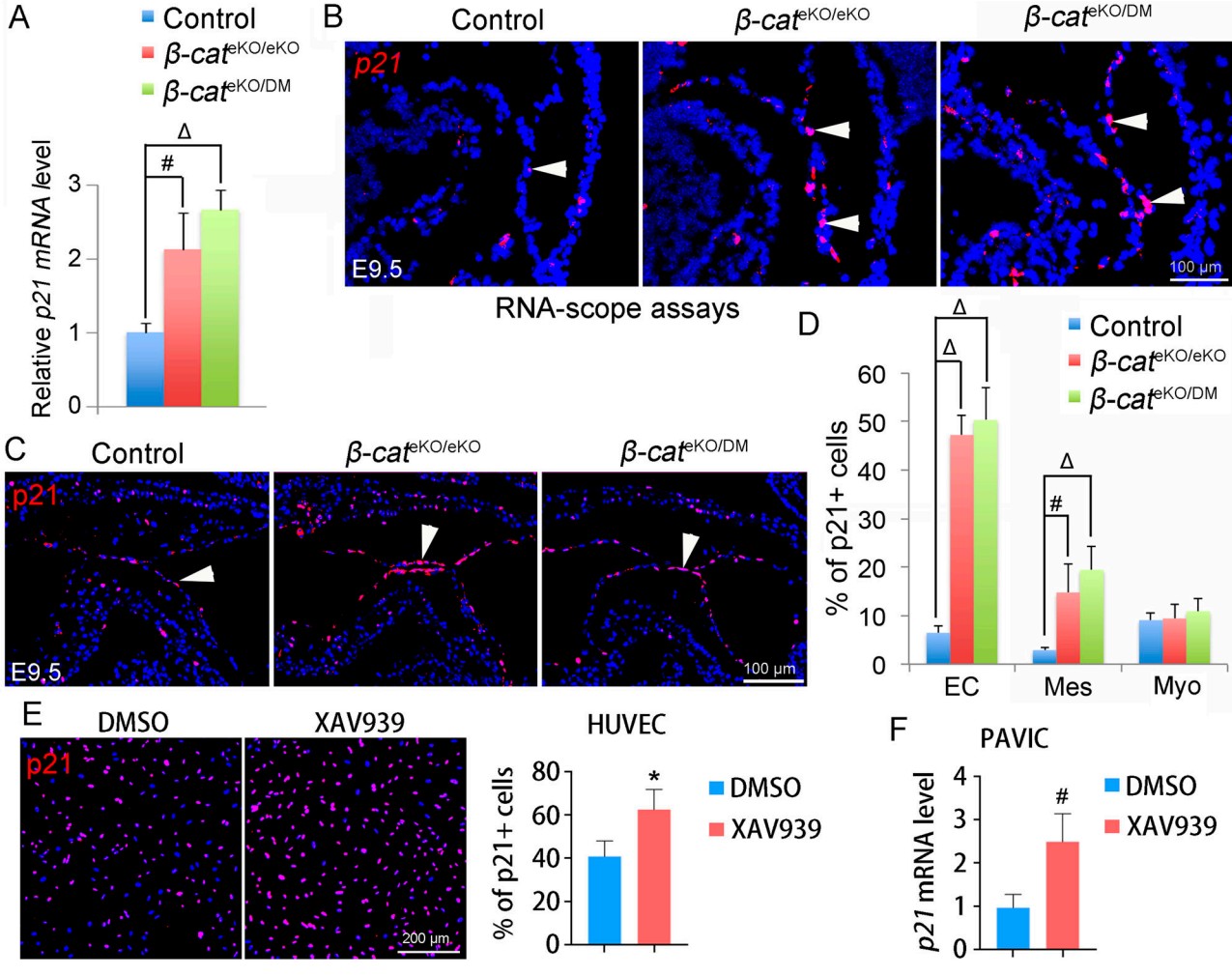

**Figure 5. β-catenin negatively regulates p21 expression.**
**(A)** AVC tissue was microdissected from E9.5 control, *β-cat*eKO/eKO, and *β-cat*eKO/DM embryos and subjected to qRT-PCR analysis of *p21* mRNA level. The expression of *p21* mRNA was normalized to that of *Gapdh*. n = 3/group. **(B)** RNAscope analysis shows *p21* mRNA expression within AVC region of E9.5 hearts. The arrowheads indicate the expression of *p21* mRNA. **(C, D)** Immunostaining for p21 (red). The percentage of p21-expressing cells (arrowhead) was quantified (D). EC, endocardial cell; Mes, mesenchymal cell; Myo, myocardial cell. n = 4/group. **(E)** HUVEC was treated with 1 μM XAV939 or DMSO for 24 h. XAV939 is an inhibitor for canonical WNT/β-catenin signaling. Immunostaining shows p21 protein expression. The percentage of p21-positive cells was quantified and presented in the bar chart on right. n = 3/group. **(F)** Pig aortic valve interstitial cells were treated with 2.5 μM XAV939 or DMSO for 24 h. The cells were subjected to qRT-PCR analysis of *p21* mRNA expression. n = 4/group. Unpaired t test and one-way ANOVA were used for the statistical calculation among two and three groups, respectively. # < 0.01; Δ < 0.001.

and cell migration during endocardial cushion formation. Similarly, β-catenin has been shown to regulate migration of neural crest cells essential for outflow septation (Kioussi et al, 2002). Together, our findings clarify that β-catenin is essential for cell proliferation and migration but dispensable for endocardial cell fate determination during early endocardial cushion formation.

β-Catenin interacts with VE-cadherin at cell membrane and mediates cell–cell adhesion. It can also shuttle to nucleus and activate target gene expression by forming a transcriptional complex with transcriptional factor TCF/LEF upon stimulation of WNT ligands (Valenta et al, 2011). By using a *β-catenin*DM allele which has the cell adhesive function but not the transcriptional activity of β-catenin, we demonstrate that β-catenin regulates cell proliferation and migration via its transcriptional function and non-transcriptional function, respectively.

Degradation of VE-cadherin is prerequisite for endocardial cells to detach from the endocardial sheet and invade the cushions (Timmerman et al, 2004). Immunostaining results showed that the protein level of VE-cadherin was increased in the cushion endocardium of β-catenin knockout embryos when compared with controls, suggesting the increased VE-cadherin protein may contribute to impaired cell migration. In contrast, the mRNA level of VE-cadherin was comparable between control and β-catenin mutants. *Snail* is known to reduce VE-cadherin expression transcriptionally and triggers the EndoMT process (Timmerman et al, 2004; Niessen et al, 2008). The protein level of SNAIL in cushion endocardium was not affected by β-catenin loss. Together, these findings suggest that β-catenin negatively regulates the protein level of VE-cadherin during endocardial cushion formation.

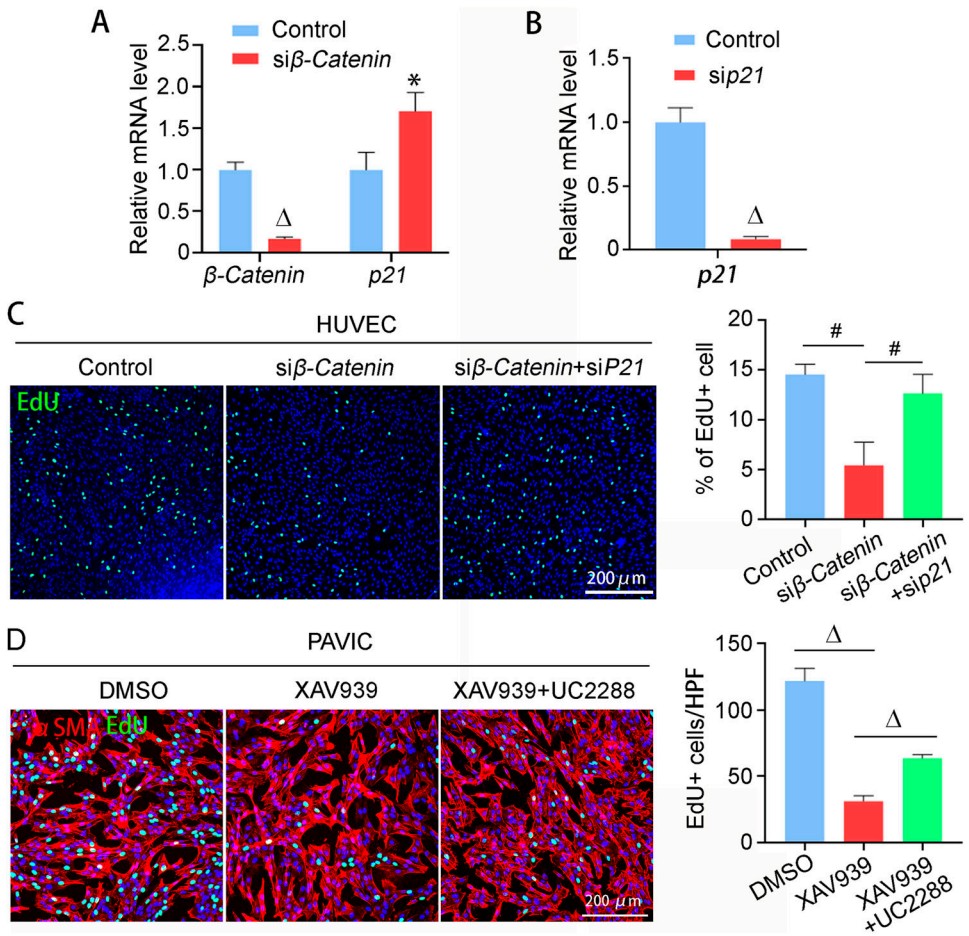

**Figure 6.  β-catenin promotes cell proliferation by suppressing p21.**
**(A, B)** HUVEC was transfected with control or gene-specific siRNA. qRT-PCR detected the mRNA expression of *β-catenin* and *p21*. n = 3/group. **(C)** HUVEC was transfected with indicated siRNA. EdU was used to label the proliferating cells. The percentage of EdU-positive cell was quantified. n = 3/group. **(D)** PAVIC were treated with XAV939 (2.5 *μ*M) and UC2288 (2.5 *μ*M). UC2288 is an inhibitor for p21. Cell proliferation was detected by EdU labeling and quantified. n = 4/group. Unpaired *t* test and one-way ANOVA were used for the statistical calculation among two and three groups, respectively. * < 0.05; # < 0.01; Δ < 0.001.

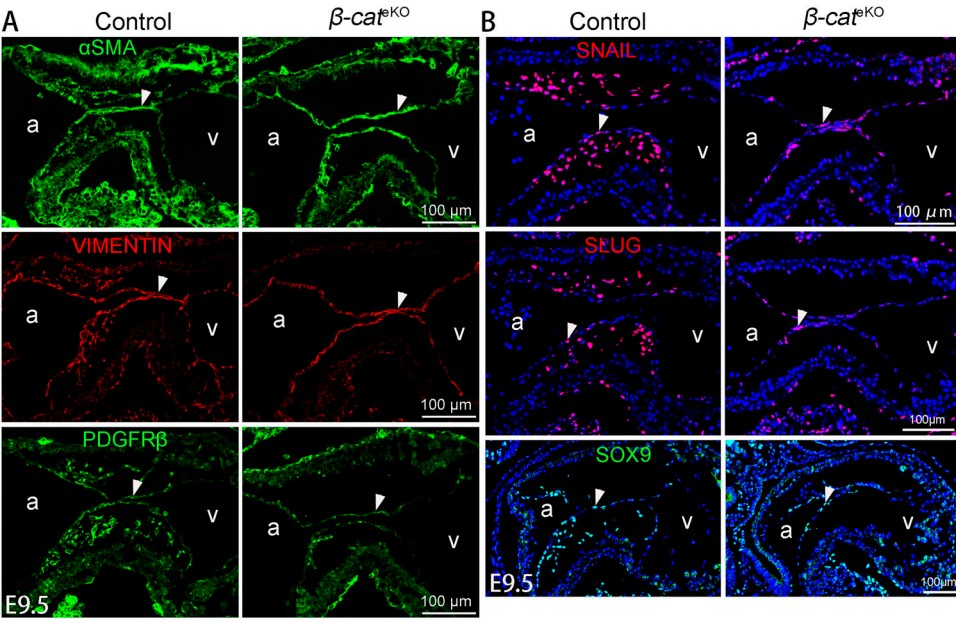

**Figure 7.  β-catenin is dispensable for endocardial-to-mesenchymal fate change.**
**(A, B)** Tissue sections of E9.5 embryos were subjected to immunostaining with antibodies for mesenchymal markers (αSMA, VIMENTIN, PDGFRβ) (A) or transcriptional factors (SNAIL, SLUG, and SOX9) involved in EndoMT process (B). The representative images show the expression of indicated markers within AVC region. The arrowheads indicate cushion endocardial cells. At least four embryos were analyzed for each staining. a, atrium; v, ventricle.

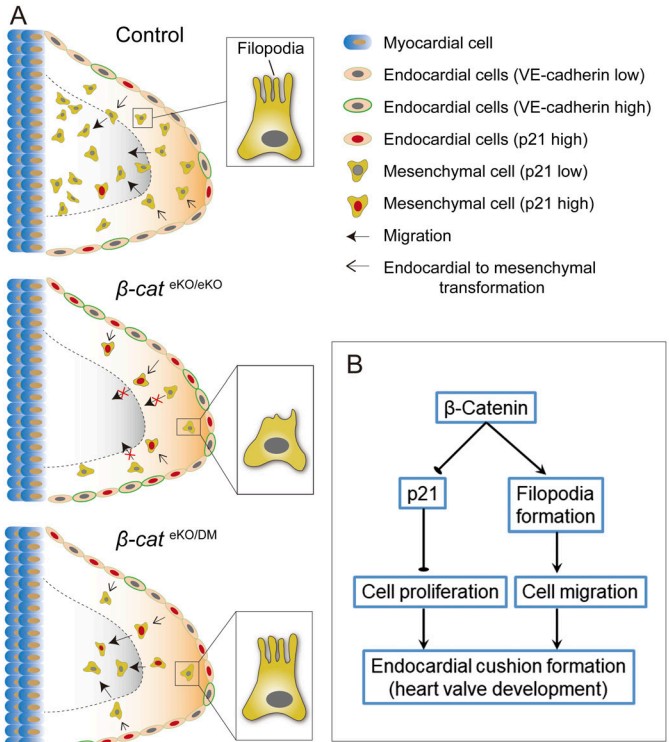

**Figure 8.  β-catenin regulates cell proliferation and migration essential for endocardial cushion formation.**
**(A)** Schematic shows the distinct cushion defects in $\beta\text{-}cat^{eKO/eKO}$ and $\beta\text{-}cat^{eKO/DM}$ embryos. At E9.5, cushion endocardial cells within AVC region of control embryos undergo EndoMT, and then the transformed mesenchymal cells delaminate from the endocardial sheet, invade into the cardiac jelly, and proliferate, generating endocardial cushions which serve as the valve primordia and eventually give rise to atrioventricular valves. In contrast, loss of β-catenin inhibits proliferation of endocardial and mesenchymal cells, leading to formation of hypocellular endocardial cushions in both $\beta\text{-}cat^{eKO/eKO}$ and $\beta\text{-}cat^{eKO/DM}$ embryos. The underlying mechanism involves up-regulation of cell cycle inhibitor p21. In addition, complete loss of β-catenin in $\beta\text{-}cat^{eKO/eKO}$ embryos impairs filopodia formation and impedes mesenchymal cell migration away from the endocardial sheet. However, selective disruption of the transcriptional function of β-catenin $\beta\text{-}cat^{eKO/DM}$ embryos does not affect cell migration. **(B)** β-catenin suppresses p21 to promote cell proliferation. On the other hand, β-catenin is essential for filopodia formation and cell migration.

We used a candidate gene approach to characterize the molecular mediators through which β-catenin regulates endocardial cushion formation. NOTCH and TGFβ/BMP are two well-known signaling pathways essential for this process. *Notch1* promotes EndoMT by reducing VE-cadherin expression via *Snail* (Timmerman et al, 2004). In zebrafish, overactivation of WNT/β-catenin signaling can induce ectopic expression of *Notch1b* and result in formation of excessive endocardial cushions (Hurlstone et al, 2003). In our study, we showed by qRT-PCR that loss of β-catenin did not affect the expression of *Notch1* and its downstream target genes (*Hey1* and *Hey2*). In line with the qRT-PCR results, immunostaining revealed that the level of active NOTCH1 protein in cushion endocardial cells was comparable between control and β-catenin mutant embryos. These results suggest that *Notch1* is not a downstream target regulated by WNT/β-catenin signaling during endocardial cushion formation in mice. This discrepancy between previous and our

study may reflect a species-specific mechanism underlying endocardial cushion formation.

Crosstalk between TGFβ/BMP and WNT/β-catenin signaling pathways has been implicated in both heart valve development and myxomatous valve disease (Chopra et al, 2017). We have previously shown that myocardial β-catenin promotes mesenchymal cell proliferation during endocardial cushion growth by inducing BMP2 expression (Wang et al, 2018). Loss of Axin2-dependent activation of WNT/β-catenin signaling augments TGFβ and BMP signaling in valve interstitial cells and leads to myxomatous valve disease (Hulin et al, 2017). In this study, we revealed by qRT-PCR and immunostaining that BMP signaling was not affected by loss of β-catenin in endocardial lineage during early endocardial cushion formation. These observations suggest that the interaction between WNT/β-catenin and BMP signaling in cardiac valves is cell context dependent. Future studies are needed to evaluate the potential effect of β-catenin loss on TGFβ signaling. Interestingly, we find that β-catenin loss dramatically upregulates p21 expression in cushion endocardial and mesenchymal cells. Moreover, p21 inhibition is able to rescue the proliferation defect caused by β-catenin deficiency in cultured HUVEC and PAVIC, suggesting that β-catenin–dependent suppression of p21 ensures cell proliferation essential for endocardial cushion formation. The mechanism behind may involve transcriptional suppression of p21 via a β-catenin/TCF-dependent manner, which has been described in HEK293 cells (Kamei et al, 2003).

In conclusion, our findings demonstrate that β-catenin is dispensable for endocardial cells to gain mesenchymal fate but essential for later cell migration and proliferation. The underlying mechanisms may involve β-catenin–mediated filopodia formation and p21 suppression. Our findings provide a new insight into the cellular and molecular mechanisms by which β-catenin controls endocardial cushion formation during early heart valve development.

# Materials and Methods

### Experimental mouse models

*Tie2^Cre* mice (Braren et al, 2006) were crossed with the floxed β-catenin mice (Brault et al, 2001) to generate conditional β-catenin knockout mice in which β-catenin was selectively deleted in the endothelium including endocardium during embryogenesis. *β-catenin^DM* mice (Valenta et al, 2011) which express a modified β-catenin protein with a single amino acid mutation (D164A) in the first armadillo repeat and a C-terminal truncation. This mutant β-catenin protein lacks any detectable transcriptional activity (signaling function) but retains the intact cell adhesive or cell–cell contact function at the cell membrane (Valenta et al, 2011). All mouse strains were maintained on the C57B6 background. Mouse experiments were performed according to the guideline of the National Institute of Health and the protocol approved by the Institutional Animal Care and Use Committee of Albert Einstein College of Medicine. Noontime on the day of detecting vaginal plugs was designated as E0.5. The embryos were isolated from the pregnant female mice which were euthanized by inhalation of

carbon dioxide gas using a euthanasia chamber. Adult mice and mouse embryos were genotyped by PCR using DNA from tail and yolk sac, respectively.

## Histology

Embryos were dissected between E9.5 and E11.5, fixed overnight at 4°C using 4% PFA in PBS, dehydrated through a serial ethanol solution, cleared with xylene, and embedded in paraffin wax. Embryos were oriented for sagittal sections and cut at 6 $\mu$m using a Leica microtome. Hematoxylin and eosin (HE) staining was performed for histology by using a standard protocol. The HE-stained tissue sections were examined and photographed using a Zeiss Axio Observer Z1 inverted microscope. Serial sections across the entire AVC cushion region were used for quantification of the number of mesenchymal cells. To analyze the distribution of mesenchymal cells in the endocardial cushions, cells located underneath the endocardium (sub-endocardium) and far away from the endocardium and underneath the myocardium (sub-myocardium) were counted, respectively. The percentage of mesenchymal cell in the sub-endocardium or the sub-myocardium was calculated and the data were presented as mean ± SD. Data from 4–6 embryos for each genotype were used for statistical calculation.

## Immunofluorescence staining

Tissue sections prepared as described above were boiled for 10 min in sodium citrate (10 mM, pH 6.0) (Vector Laboratories) to retrieve antigens and then blocked with 5% normal horse serum in PBS before being incubated with primary and secondary antibodies (Table S1). The signal was amplified using the TSA Plus Cyanine 3 System (Perkin Elmer) as noted for some antibodies. The stained tissue sections were photographed using a Leica confocal microscope.

## EdU staining

Cell proliferation was determined by the EdU assay. 2 h before collecting embryos, pregnant female mice at E9.5 were administrated with EdU (Life Technology) through intraperitoneal injection at a concentration of 100 mg/kg. After the 2 h chasing, embryos were fixed in 4% PFA at 4°C for 1 h, soaked in 15% and 30% sucrose sequentially, and embedded in OCT compounds in the sagittal orientation. The tissues were cut at 8 $\mu$m thickness and mounted on the positive charged slides. Tissue sections were then fixed in cold ethanol and acetone (1:1) solution for 5 min and stored at –80°C. Before staining, tissue sections were air dried for 45 min. Serial sections crossing the entire cushion region were first stained with PECAM1 antibody followed by EdU staining using an EdU imaging Kit (Life Technology) and counterstained with DAPI. The stained sections were photographed using a Leica confocal microscope. The positive EdU labeling was quantified for the endocardium, mesenchyme, or myocardium using ImageJ software. At least three embryos were analyzed for each genotype.

## RNA extraction and qRT-PCR

The AVC cushion tissues were microdissected from E9.5 embryos and subjected to isolation of total RNAs using TRIzol (Invitrogen). Tissues from 2–3 embryos were pooled as one sample. First-strand cDNA was synthesized using the Superscript II Reverse Transcriptase Kit (Invitrogen). qRT-PCR was carried out using the Power SYBR Green PCR Master Mix (ABI). Gene-specific primers were used for genes of interest (Table S2). The relative level of gene expression was normalized to an internal control (level of *Gapdh*) and calculated using the $2^{-\Delta\Delta CT}$ method. Three samples per genotype were analyzed, and the mean relative expression of each gene between groups was used for statistical analysis.

## RNAscope

E9.5 embryos were collected and processed for frozen sections. mRNA expression pattern of genes was detected by using an RNAscope 2.5 HD Reagent Red Kit (ACD BIO) according to the manufacturer's instructions. The images were collected using a Leica confocal microscope.

## In vitro EndoMT assay

EndoMT assay was performed as described previously (Lencinas et al, 2011). AVC tissues were microdissected out from E9.5 embryos and cultured on the rat tail collagen gel in four-well plates. Explants were cultured for 48 h and stained with phalloidin to label the F-actin, whereas nuclei were stained by DAPI. Cells that migrated away from the AVC explants and invaded into the gel were counted. To track the cell migration, time-lapse images were collected and subjected to cell tracking analysis using DiPer as described previously (Gorelik & Gautreau, 2014).

## Cell culture

Primary HUVECs and PAVIC were cultured in endocardial cell medium (ScienCell Research Laboratories) and DMEM, respectively. To block the canonical WNT signaling, XAV939 were added to the culture media at a final concentration of 1 $\mu$M for HUVEC or 2.5 $\mu$M for PAVIC. PAVIC was treated with 2.5 $\mu$M of UC2288 to inhibit p21. To knockdown the expression of *Ctnnb1* (*β-catenin*) and *Cdkn1A* (*p21*), the cells were transfected with gene-specific siRNA, whereas the scramble siRNA were introduced simultaneously as controls. The knockdown efficiency was evaluated using qRT-PCR. EdU was added into the culture media with a final concentration of 10 $\mu$M. 2 h later, the cells were processed for EdU staining as described above. Immunostaining was performed to detect the expression level of p21 protein.

## Statistical analysis

Statistical analysis was performed using GraphPad Prism. All data were presented as mean ± SEM. *t* test was used for comparisons between two groups. One-way ANOVA was used for comparisons among three groups. Probability value < 0.05 was considered as significant.

# Life Science Alliance

## Supplementary Information

## Acknowledgements

We think Dr. Tomas Valenta and Dr. Konrad Basler for providing *β-catenin*[DM] mice. Y Wang is supported by the grants from the National Natural Science Foundation of China (81970266), the National Key R&D Program of China (2019YFA0802300), and Department of Human and Social Affairs of Shaanxi Province in China (2019001). B Zhou is supported by the grants from the National Heart, Lung, and Blood Institute (R01HL133120, R01HL148128, R01HL157347, and R01HL159515).

### Author Contributions

H Liu: conceptualization, investigation, and writing—original draft.
P Lu: conceptualization, investigation, and writing—review and editing.
S He, Y Luo, and Y Fang: investigation.
S Benkaci: formal analysis and investigation.
B Wu: supervision and investigation.
Y Wang: conceptualization, supervision, investigation, and writing—original draft, review, and editing.
B Zhou: conceptualization, supervision, and writing—original draft, review, and editing.

### Conflict of Interest Statement

The authors declare that they have no conflict of interest.

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
