## [Reviewer comments · Life Science Alliance]

Life Science Alliance

β -catenin regulates endocardial cushion growth by suppressing p21

Huahua Liu, Pengfei Lu, Shan He, Yuru Luo, Yuan Fang, sonia benkaci, bingruo Wu, Yidong Wang, and Bin Zhou
DOI: <https://doi.org/10.26508/lsa.202302163>

Corresponding author(s): Bin Zhou, Albert Einstein College of Medicine and Yidong Wang, First Affiliated Hospital of Xi'an Jiaotong University

Review Timeline:

Submission Date:	2023-05-15
Editorial Decision:	2023-06-05
Revision Received:	2023-06-16
Editorial Decision:	2023-06-20
Revision Received:	2023-06-21
Accepted:	2023-06-22

Scientific Editor: Novella Guidi

Transaction Report:

June 5, 2023

Re: Life Science Alliance manuscript #LSA-2023-02163

Dr. Bin Zhou
Albert Einstein College of Medicine
1301 Morris Park Ave
Bronx 10461

Dear Dr. Zhou,

Thank you for submitting your manuscript entitled " β -catenin regulates endocardial cushion growth by suppressing p21" to Life Science Alliance. The manuscript was assessed by expert reviewers, whose comments are appended to this letter. We invite you to submit a revised manuscript addressing the Reviewer comments.

Thank you for this interesting contribution to Life Science Alliance. We are looking forward to receiving your revised manuscript.

Sincerely,

B. MANUSCRIPT ORGANIZATION AND FORMATTING:

Reviewer #1 (Comments to the Authors (Required)):

In this ms, this group of researchers studied the role of beta-catenin in regulating cardiac valve formation. The ms was very well written and sufficient background information was provided to support the rationale and significance of the research. Experiments were well designed and data are of high quality. The conclusion is highly convincing based on data provided.

I have only one minor concern.

1. The cre line used should also inactivate target genes in the OFT region. It will be great if they can add some description regarding the phenotype in the OFT region either in Results or Discussion.

Reviewer #2 (Comments to the Authors (Required)):

Liu et al examined the functions and mechanisms underneath of how beta-catenin regulates endocardial cushion growth via different genetic tools including tissue specific beta-catenin deletion and a mouse line that can generate a truncated beta-catenin. The major conclusion is that β -catenin is essential for cell proliferation and migration but dispensable for endocardial cells to gain mesenchymal fate during endocardial cushion formation. Mechanistically, β -catenin promotes cell proliferation by suppressing p21 and cell migration by promoting the formation of microspikes or filopodia.

The study is well designed and well executed, and the data is convincing.

The minor question is that Beta-catenin ko does not affect Notch signaling in this study, but overactivation of Wnt/beta-catenin induced ectopic expression of Notch1b in zebrafish. Can the authors comment on it?

Reviewer #3 (Comments to the Authors (Required)):

β -catenin regulates endocardial cushion growth by suppressing p21

Liu et al demonstrated that β -catenin is essential for cell proliferation and migration but dispensable for endocardial cells to gain mesenchymal fate during endocardial cushion formation. Mechanistically, β -catenin promotes cell proliferation by suppressing p21.

This is a very important manuscript which revisits the cushion formation. Given the enormous published literature on Wnt/ β catenin in cushion formation and cushion remodeling, it is very easy to dismiss this study by saying that this manuscript is somewhat incremental. This manuscript simply dismiss this unfounded misconception in the scientific community that we know everything how well-studied major signaling pathways (Wnt, TGF β , BMP) function in heart development, and therefore, there is no further need to investigate these signaling pathways. This manuscript, by re-investigating and carefully adding new experiments, elegantly showed the novel unanticipated role of Wnt/ β catenin in cushion mesenchymal proliferation and remodeling. This important finding would have been simply missed if scientists have no incentives or encouragement for studying these well-known pathways involved in heart development. The findings from this manuscript advance our knowledge about the potential role of β -catenin in the etiology of congenital heart defects.

Minor suggestion:

- In the Discussion, the authors have discussed the connection between Wnt/ β catenin to Notch and BMP signaling in cushion formation. Given that TGF β ligands play critical role in cushion formation and remodeling (another unappreciated area of investigation in the field), it would be valuable to include TGF β ligands/signaling in the discussion. Please review: Increased canonical WNT/ β -catenin signalling and myxomatous valve disease (PMID: 28069697). This review describes potential integration of Wnt/ β catenin, BMP, and TGF β in heart valve development and homeostasis.

We thank the editor and reviewers for your time and insightful comments. We have revised the manuscript according to your suggestions by including the OFT data and additional discussions on the interactions among WNT/ β -catenin, TGF β /BMP, and NOTCH signaling pathways. Below are our point-to-point responses. Changes in the revised manuscript are highlighted in purple font.

Reviewer #1 (Comments to the Authors (Required)):

In this ms, this group of researchers studied the role of beta-catenin in regulating cardiac valve formation. The ms was very well written and sufficient background information was provided to support the rationale and significance of the research. Experiments were well designed and data are of high quality. The conclusion is highly convincing based on data provided.

I have only one minor concern.

1. The cre line used should also inactivate target genes in the OFT region. It will be great if they can add some description regarding the phenotype in the OFT region either in Results or Discussion.

Response: We appreciate the reviewer for the encouraging comments and have added discussions on the phenotype in the OFT region in the Result section (please see page 5, lines 110-113). Tie2^{Cre} also deletes genes at the OFT region where EndoMT starts around E9.5, slightly later than the beginning time of EndoMT in the AVC region (Kisanuki, Hammer et al., 2001). We have included a new Figure S2 to show that there were no mesenchymal cells in the OFT cushions of either control or beta-catenin mutant embryos at E9.5.

Reviewer #2 (Comments to the Authors (Required)):

Liu et al examined the functions and mechanisms underneath of how beta-catenin regulates endocardial cushion growth via different genetic tools including tissue specific beta-catenin deletion and a mouse line that can generate a truncated beta-catenin. The major conclusion is that β -catenin is essential for cell proliferation and migration but dispensable for endocardial cells to gain mesenchymal fate during endocardial cushion formation. Mechanistically, β -catenin promotes cell proliferation by suppressing p21 and cell migration by promoting the formation of microspikes or filopodia. The study is well designed and well executed, and the data is convincing.

The minor question is that Beta-catenin ko does not affect Notch signaling in this study, but overactivation of Wnt/beta-catenin induced ectopic expression of Notch1b in zebrafish. Can the authors comment on it?

Response: We thank the reviewer for the encouraging comments and have included discussions about the seemingly discrepancy between previous and our studies in the revised manuscript (please see page 10, lines 287-296). Briefly, *Notch1* promotes EndoMT by reducing VE-cadherin expression via *Snail* (Timmerman et al., 2004). In zebrafish, over-activation of WNT/ β -catenin signaling can induce ectopic expression of *Notch1b* and result in hypercellular endocardial cushions (Hurlstone et al., 2003). However, our qPCR results showed that loss of β -catenin did not affect the expression of *Notch1* as well as its downstream target genes (Hey1 and Hey2). In line with the qPCR results, immunostaining revealed that the level of active NOTCH1 protein in the cushion endocardial cells was comparable between control and β -catenin mutant embryos. These results suggest that *Notch1* is not a downstream target of the WNT/ β -catenin signaling during endocardial cushion formation in mice. This discrepancy between previous and our study may reflect a species-specific mechanism underlying endocardial cushion formation.

Reviewer #3 (Comments to the Authors (Required)):

β -catenin regulates endocardial cushion growth by suppressing p21

Liu et al demonstrated that β -catenin is essential for cell proliferation and migration but dispensable for endocardial cells to gain mesenchymal fate during endocardial cushion formation. Mechanistically, β -catenin promotes cell proliferation by suppressing p21.

This is a very important manuscript which revisits the cushion formation. Given the enormous published literature on Wnt/ β catenin in cushion formation and cushion remodeling, it is very easy to dismiss this study by saying that this manuscript is somewhat incremental. This manuscript simply dismiss this unfounded misconception in the scientific community that we know everything how well-studied major signaling pathways (Wnt, TGF β , BMP) function in heart development, and therefore, there is no further need to investigate these signaling pathways. This manuscript, by re-investigating and carefully adding new experiments, elegantly showed the novel unanticipated role of Wnt/ β catenin in cushion mesenchymal proliferation and remodeling. This important finding would have been simply missed if scientists have no incentives or encouragement for studying these well-known pathways involved in heart development. The findings from this manuscript advance our knowledge about the potential role of β -catenin in the etiology of congenital heart defects.

Minor suggestion:

- In the Discussion, the authors have discussed the connection between Wnt/ β catenin to Notch and BMP signaling in cushion formation. Given that TGF β ligands play critical role in cushion formation and remodeling (another unappreciated area of investigation in the field), it would be valuable to include TGF β ligands/signaling in the discussion. Please review: Increased canonical WNT/ β -catenin signalling and myxomatous valve disease (PMID: 28069697). This review describes potential

integration of Wnt/ β catenin, BMP, and TGF β in heart valve development and homeostasis.

Response: We thank the reviewer for positively commenting on our manuscript and the insightful suggestion. We now include the potent TGF β signaling and cite the reference (PMID: 28069697) in our discussions on the potential collaborative or integrated signaling events in play during endocardial cushion formation (please see page 10, lines 298-307). Briefly, crosstalk between TGF β /BMP and WNT/ β -catenin signaling pathways has been implicated in both heart valve development and myxomatous valve disease (Chopra, Al-Sammarraie et al., 2017). We have previously shown that myocardial β -catenin promotes mesenchymal cell proliferation during endocardial cushion growth by inducing BMP2 expression (Wang et al., 2018). Loss of Axin2-dependent activation of WNT/ β -catenin signaling augments TGF β and BMP signaling in valve interstitial cells and leads to myxomatous valve disease (Hulin et al., 2017). In this study, we revealed by qPCR and immunostaining that BMP signaling was not affected by loss of β -catenin in endocardial lineage during early endocardial cushion formation. These observations suggest that the interaction between WNT/ β -catenin and BMP signaling in cardiac valves is cell context dependent. Future studies are needed to evaluate the potential effect of β -catenin loss on TGF β signaling.

June 20, 2023

RE: Life Science Alliance Manuscript #LSA-2023-02163R

Dr. Bin Zhou
Albert Einstein College of Medicine
1301 Morris Park Ave
Bronx 10461

Dear Dr. Zhou,

Thank you for submitting your revised manuscript entitled " β -catenin regulates endocardial cushion growth by suppressing p21". We would be happy to publish your paper in Life Science Alliance pending final revisions necessary to meet our formatting guidelines.

- please add ORCID ID for the secondary corresponding author--they should have received instructions on how to do so
- please add the Twitter handle of your host institute/organization as well as your own or/and one of the authors in our system
- please use the [10 author names et al.] format in your references (i.e., limit the author names to the first 10)
- please upload figures only individually and provide their legends after the references section in the manuscript text
- please add your main, supplementary figure, table, and video legends to the main manuscript text after the references section
- please add callouts for Figure 8A-B to your main manuscript text

A. FINAL FILES:

B. MANUSCRIPT ORGANIZATION AND FORMATTING:

Sincerely,

June 22, 2023

RE: Life Science Alliance Manuscript #LSA-2023-02163RR

Dr. Bin Zhou
Albert Einstein College of Medicine
1301 Morris Park Ave
Bronx 10461

Dear Dr. Zhou,

Thank you for submitting your Research Article entitled " β -catenin regulates endocardial cushion growth by suppressing p21". It is a pleasure to let you know that your manuscript is now accepted for publication in Life Science Alliance. Congratulations on this interesting work.

DISTRIBUTION OF MATERIALS:

Again, congratulations on a very nice paper. I hope you found the review process to be constructive and are pleased with how the manuscript was handled editorially. We look forward to future exciting submissions from your lab.

Sincerely,
